# Green Preparation of Lightweight, High-Strength Cellulose-Based Foam and Evaluation of Its Adsorption Properties

**DOI:** 10.3390/polym15081879

**Published:** 2023-04-14

**Authors:** Yongxing Zhou, Wenbo Yin, Yuliang Guo, Chenni Qin, Yizheng Qin, Yang Liu

**Affiliations:** 1College of Light Industry and Food Engineering, Guangxi University, Nanning 530004, China; 2005170220@st.gxu.edu.cn (Y.Z.);; 2Shandong Institute of Standardization, Jinan 250000, China; 3Guangxi Key Laboratory of Clean Pulp and Paper and Pollution Control, Guangxi University, Nanning 530004, China

**Keywords:** cellulose-based foam, gelatin, adsorption

## Abstract

In recent years, the application scope of most cellulose-based foams is limited due to their low adsorbability and poor recyclability. In this study, a green solvent is used to extract and dissolve cellulose, and the structural stability of the solid foam is enhanced by adding a secondary liquid via the capillary foam technology, and the strength of the solid foam is improved. In addition, the effects of the addition of different gelatin concentrations on the micro-morphology, crystal structure, mechanical properties, adsorption, and recyclability of the cellulose-based foam are investigated. The results show that the cellulose-based foam structure becomes compact, the crystallinity is decreased, the disorder is increased, and the mechanical properties are improved, but its circulation capacity is decreased. When the volume fraction of gelatin is 2.4%, the mechanical properties of foam are the best. The stress of the foam is 55.746 kPa at 60% deformation, and the adsorption capacity reaches 57.061 g/g. The results can serve as a reference for the preparation of highly stable cellulose-based solid foams with excellent adsorption properties.

## 1. Introduction

Crude oil leakage and industrial emissions have led to high levels of pollution in the ecological environment. Recently, bio-waste and natural resources have been considered for developing green synthetic nanomaterials/nanoparticles. These green nanoparticles may be employed as a viable alternative to current methods of pollution remediation since they are cheap, stable, safe, and environmentally benign [1]. Among them, the method of adsorbing oil-water by materials with adsorption properties has been widely concerned in pollution control. Cellulose-based foam has attracted much attention as an adsorbent and is widely reported in the literature [2,3,4,5] as one of the sewage treatment materials [3,5,6,7,8,9,10,11,12] with the highest potential owing to its light weight, low density, high porosity, and high liquid holdup [13]. However, chemical modifications can often cause serious damage to the network structure of the foam and reduce its service life. Therefore, it is crucial to find an oil-absorbing cellulose-based foam with a stable structure and good recyclability performance.

Cellulose is the primary component of lignocellulosic biomass which can serve as a plentiful source of carbohydrates for the production of numerous high-demand chemicals [14]. The process of preparing cellulose-based foams requires the extraction and dissolution of cellulose by a green method. Cellulose extraction techniques include mechanical extraction by homogenization, mechanical isolation by steam explosion, defibrillation by high-intensity ultrasonication, extraction by electrospinning technique, extraction and dissolution of cellulose using ionic liquids, etc. [15]. The traditional cellulose extraction and dissolution process is not in line with the current trend of energy saving and environmental protection. Therefore, in order to fully utilize cellulose resources, it is necessary to develop “green” cellulose extraction methods and suitable cellulose dissolution pathways. As neoteric green solvents, ionic liquids (ILs) refer to a specific class of molten salts which are liquids at temperatures of 100 °C [16,17]. In recent years, many researchers have investigated the dissolution of cellulose in green solvents(ILs), Froschauer et al. [18] separated cellulose and hemicellulose from wood pulp by 1--ethyl--3--methylimidazolium acetate [C2mim][Ac]/Cosolvent (water, ethanol, or acetone) systems. Zhuo et al. [19] synthesized SO_3_H^−^ functionalized acidic ionic liquids used as catalysts for the hydrolysis of cellulose in [C4mim][Cl] and so on. The dissolution mechanism of cellulose in ILs [20] involves the oxygen and hydrogen atoms of cellulose-OH in the formation of electron donor-electron acceptor (EDA) complexes which interact with the ionic liquid. The cations in ionic liquid solvents act as the electron acceptor center and anion as the electron-donor center. The two centers must be located close enough in space to permit the interactions and to permit the EDA complexes to form. Upon interaction of the cellulose-OH and the ionic liquid, the oxygen and hydrogen atoms from hydroxyl groups are separated, resulting in the opening of the hydrogen bonds between molecular chains of the cellulose, and finally, the cellulose dissolves [21].

Increasing the stability of wet foam is considered an effective method to prevent structural damage of the cellulose-based foam network. Foam is a complex metastable system that is influenced by many factors. According to the Reynolds equation [22], foam decay is related to the liquid film thickness, solution viscosity, and the size of bubbles [23,24]. Decay is often attributed to the foam thinning and rupture caused by the liquid film drainage, which is controlled by the viscoelasticity of the liquid film. Therefore, the viscoelasticity of the liquid film is a very important factor influencing the stability of the foam. In general, adding a surfactant will play a positive role in improving the stability of the foam. The foam forming of cellulose-based materials is influenced by the surfactant type and dosage [25]. During foam formation, the surfactants are arranged in an ordered manner on the liquid film surface, wherein the hydrophilic group is directed toward the water and the hydrophobic group is directed toward the air [26]. When the concentration of surfactants reaches the critical micelle concentration, the interface adsorption is saturated and micelles are formed. On one hand, the stabilization mechanism reduces the liquid discharge power by reducing the Laplace pressure; on the other hand, it imparts the liquid film with a certain degree of Gibbs film elasticity. When the liquid film is impacted, local deformation occurs. Consequently, the adsorption density of the surfactant in this area decreases, and a surface tension gradient is formed within the adjacent area. Therefore, the liquid in the liquid film flows from the low-surface-tension area to the high-surface-tension area [22], thereby preventing deformation (thinning) and rupture of the liquid film. Moreover, the increase in solution viscosity plays a secondary role in the stability of the foam. According to the literature [27], within the range of foaming ability, higher solution viscosity leads to a more stable foam. Chen et al. [28] studied the stability and rheological properties of foams prepared from surfactants and clay particle dispersion. The author thus confirmed that an increase in solution viscosity can indeed enhance the stability of the foam. In the case of cellulose-based wet foams, the size of the foam directly determines the pore size of the subsequent cellulose porous foam. Compared to the pure water-vapor two-phase foam, the cellulose-based wet foam is subjected to a much stronger capillary force and external force during the curing process, which renders the foam extremely unstable. Therefore, the method of formation of a dense network structure is important for the preparation of stable cellulose-based wet foams. In a previous study [29], the cationic character of a cationic polyacrylamide (CPAM) was successfully exploited to capture the negatively charged free nanofibers (NFC) in an aqueous solution. Therefore, during the preparation of a cellulose-based stable wet foam, the electrostatic force between CPAM and NFC can be fully utilized. The NFC is used as a patch, and the local adsorption of CPAM on the large cellulose is used as a bonding point to fill the NFC in the gap of the cellulose skeleton [30], and this is called the “patch bridge” mechanism. By changing the amount of NFC added, the spatial distance between the cellulose is changed, thus changing the density of the cellulose network and thereby improving that stability of the cellulose-based wet foam. Furthermore, the factors affecting the stability of the cellulose-based wet foams also include the foaming agent concentration, cellulose concentration, stirring speed, etc. [31,32,33,34].

In addition, gelatin is generally utilized for structure changes and increased porosity [35]. To improve the resistance of the cellulose network structure, a capillary foam can be formed by adding a second liquid (gelatin) to increase the yield stress, thereby reinforcing the cellulose skeleton. According to literature reports [36,37], the increase in the yield stress can be explained through two possibilities: (1) formation of physical bonds between the particles after adding the secondary liquid and (2) hydrogen bonding between the cellulose and the gelatin and the increase in the volume of the single-capillary bridge connecting the cellulose [38]. Moreover, it was found that, when an external force was applied, the stress can be transferred in the polymer network structure and dispersed [39]. Hydrogen bonds are easy to design and have dynamic reversibility under external stimulation [40]. With the addition of the second liquid, the cellulose gradually forms a network that is connected by liquid bridges across the bubbles, with a significant increase in the yield stress [37]. When the critical volume fraction is reached, the number of liquid bridges reaches saturation, indicating an insignificant increase in the yield stress. However, when the volume fraction increases beyond the critical value, the yield stress increases rapidly. At this time, the increase in the volume of capillary liquid bridges between the cellulose generates a strong yield stress [37]. The results show that the second liquid is imperative for the formation of a network of particles interconnected by liquid bridges, which then changes into a gel-like substance with high elasticity [41]. This transformation significantly increases the yield stress and viscosity, which are higher than those of an ordinary cellulose foam. Therefore, the cellulose skeleton is enhanced, and the resistance ability of the cellulose network structure is improved.

In this study, gelatin is used as the second liquid, with capillary foam technology to improve the stability of the cellulose skeleton. The effects of gelatin concentration in the foam on the microstructure, crystal structure, and mechanical properties of the cellulose-based foam are studied. Furthermore, the adsorption and cycle performance of the cellulose-based foam with different gelatin concentrations are analyzed by adsorption cycle experiments which are carried out by using the traditional extrusion–drying–absorption cycle method. However, there is still a lack of clear analysis of the adsorption cycle times, and we hope that some researchers will conduct detailed and clear research in the future.

## 2. Materials and Methods

### 2.1. Materials

Bagasse fiber (BF) was obtained from Guangxi Guitang Group. Nanofibrillated cellulose (NFC) was purchased from Tianjin Damao Chemical Reagent Factory. Sodium lauryl sulfate (SDS), 1-tetradecyl alcohol (TDA), and gelatin were purchased from Shanghai Aladdin Biochemical Technology Co., Ltd. (Shanghai, China). Gum arabic (GAC) was purchased from Tianjin Damao Chemical Reagent Factory. All reagents were used as is, unless otherwise stated.

### 2.2. Experimental Devices

The experimental devices used are as follows: high shear dispersion emulsifier (FM 200, Frug Fluid Machinery Co., Ltd., Foshan, China), high-speed disperser (Ultra-Turrax, Aika, Germany), thermostat water bath cauldron (DF-101S, Bonsai Instruments Co., Ltd., Shanghai, China), freeze-dryer (E35 A-Pro, Shanghai Qiaofeng Industrial Co., Ltd., Shanghai, China), field emission scanning electron microscope (Gemini500, Zeiss Instruments, Germany). X-ray computed tomography (GE Vtomx, GE, the US), automatic mercury injection apparatus (AutoPore IV 9500, McMurray, the US), X-ray diffractometer (Rigaku D/MAX 2500V, Neo-Confucianism, Japan), and universal tensile testing machine (LS1, AMETEK, the US).

### 2.3. Preparation of Cellulose-Based Foam

The cellulose-based composites are developed by dispersing the stable aqueous suspensions of cellulose mostly in hydrosoluble or hydrodispersible or latex-form polymers [15], and the cellulose-based foam is prepared by the freezing method. At first, the study uses SDS as a green solvent, and 0.015 g/mL SDS, 0.1 g/mL GAC, and 0.015 g/mL TDA solutions are used as foaming solutions. The solution is heated in a water bath at 60 °C for 5 min and then taken out when its appearance changes from turbid into clear light blue. The solution is cooled to room temperature and then transferred into a blast furnace with a dry weight accounting for 1.8% of the whole system. NFC is then dispersed by an emulsifier and added to 20% of BF dry weight. The mixture is stirred at a low speed of 800 rpm for 10 min and then foamed at a high speed of 2000 rpm for 15 min. After foaming, a small amount of CPAM is added dropwise while stirring constantly. After stirring for 5 min, the gelatin is added dropwise until the volume accounts for 0%, 1.2%, 1.6%, 2.4%, and 3.2% of the total weight of the system. The samples are named NPB, NPGB-1.2, NPGB-1.6, NPGB-2.4, and NPGB-3.2, respectively. Then, the solution is poured into a round mold of a 10 cm diameter and frozen overnight in a refrigerator at −20 °C. After the above series of operations, the foam is dried in a freeze dryer at −65 °C for 36 h to obtain a solid foam containing different concentrations of gelatin.

### 2.4. Field-Emission Scanning Electron Microscopy (FESEM)

The microstructure of the cellulose-based foam is analyzed by FESEM (Gemini500, Giebelstadt, Germany). Before testing, the foam is quenched in liquid nitrogen and adhered to the stage with a conductive adhesive. All samples are gold sprayed under vacuum for 60 s, and the samples are tested at 10 kV. The porosity is calculated according to the Formula (1): [42,43]
(1)$$Porosity=\left(1-\frac{\rho_{b}}{\rho_{s}}\right)\times 100{\%}^{\;}$$
(2)$$\rho_{b}=\frac{m}{\upsilon }$$

*ρ_b_* is the bulk density of cellulose foam, *ρ_s_* is the skeleton density of cellulose, which is 1.5 g/cm^3^, and *ρ_b_* is the density of the foam material, calculated according to Formula (2) [44].

### 2.5. X-ray Computed Tomography (Micro-CT)

The cellulose-based solid-state foam is cut into small pieces of 1 × 1 × 1 cm and scanned at the source voltage of 20 kV X-ray tube, with an image resolution of 2 μm.

### 2.6. Pore-Size Distribution

The pore-size distribution of the foam is measured by an automatic mercury injection apparatus. The cellulose-based foam is cut into pieces of 1.5 cm × 1 cm× 1 cm and dried overnight in an oven at 60 °C before the measurement. The measured contact angle is 130°, and the pressure ranged from 0.10 to 61,000 psia.

### 2.7. X-ray Diffraction (XRD)

The crystal structure of the foam cellulose is analyzed by XRD. The samples are cut into 1 mm thick slices for testing. Cu-K*α* rays with a wavelength of *λ* = 0.154 nm are used for scanning analysis. The voltage is 40 kV, the current is 30 mA, scanning diffraction angle “2*θ*” is within the range 5–50°, and scanning speed is 3 °/min. Combined with the analysis software JADE, the crystallinity index (*CrI*) is calculated according to Formula (3) [45].
(3)$$CrI=\frac{I_{002}-I_{am}}{I_{002}}\times 100\%$$

The diffraction peak intensity obtained when *I*_002_ is at 2*θ* = 22.5° is the diffraction intensity attributed to the crystalline region, and the diffraction peak intensity obtained when *I_am_* is at 2*θ* = 16.5° is the diffraction intensity attributed to the amorphous region.

### 2.8. Compression Testing

The mechanical strength of the foams is analyzed using a universal tensile tester (LS1, USA). The compressive properties of cellulose-based foams are tested according to GB/T 8813-2008 “determination of compression properties of rigid foams”. For this test, cylindrical specimens with a height of 15 mm and a diameter of 20 mm are prepared using a die with a height of 17 mm and a diameter of 22.5 mm. Before the test, all samples are placed in a vacuum drying oven and kept at 25 °C for 12 h. The test is carried out at 10 mm/min until the material strain reaches 80%.

### 2.9. Adsorption Cycle Capacity Test

Two common organic solvents and six different oils are used as adsorbents. The cellulose-based foam is cut into small pieces of 20 mm× 20 mm× 10 mm, and the initial weight of each sample is denoted by *M*_0_. In the adsorption experiment, the sample is immersed in different adsorption solvents until reaching adsorption saturation, and then the excess surface liquid is scraped off with paper. The weight after adsorption is denoted by *M_t_*, and the liquid adsorption *Q*_s_ is calculated as follows (4): [46]
(4)$$Q_{s}=\frac{M_{t}-M_{0}}{M_{0}}$$

The desorption step involves simply squeezing the adsorbed samples, washing them with ethanol and soaking them overnight and finally drying them in a vacuum oven at 60 °C for 10 h. The whole above process is referred to as a cycle.

## 3. Results and Discussion

### 3.1. Morphological Characteristics of the Cellulose-Based Foam

The cellulose-based foam is prepared by the foaming–molding method, and its cell structure is directly related to the strength of the network structure of the cellulose-based stable wet foam before curing. Therefore, under the conditions of different gelatin concentrations, the final cellulose-based foam has obvious differences in appearance and morphology.

Figure 1a shows an actual foam picture before (NPB) and after adding gelatin (NPGB). It can be seen from the figure that the foam of the former is softer than that of the latter and has a poorer performance, which makes it difficult to perform the relevant characterization by scanning electron microscopy. However, by comparing the X-ray diffraction patterns of the two (Figure 1b), it is found that the foam structure after adding gelatin becomes significantly denser. In the figure, black represents the pores, white represents the cellulose, and the brighter the color, the denser is the cellulose or the cellulose distribution at that location. As revealed in the figure, the white highlighted areas of the foam are banded together without gelatin, and the distance between the bands are large. This indicates a phenomenon where the cellulose gathers in piles, and the skeleton structure in the foam is hollow. With the addition of gelatin, the brightness of the white-banded area decreases, and the area itself also shrinks. It is evenly distributed in the form of bright white dots. The above phenomenon indicates that the cellulose is uniformly dispersed and its structure is compact. The results show that the high yield stress is imparted to the cellulose skeleton via the capillary foam technology. Therefore, it can successfully maintain the network structure of the foam, which can also be reflected by the appearance of the foam (Figure 1a). NPB has a soft structure, while NPGB is smooth and delicate with a firm texture.

Figure 1c shows the surface (top) and cross-sectional (bottom) SEM images of cellulose-based foams with different gelatin concentrations. It can be seen from the surface SEM images that all the samples maintain a good cell structure in the transverse direction. However, in the machine direction, when the volume fraction of gelatin is 1.2%, the foam appears hollow, which is known as the collapse phenomenon. This is because although the gelatin provides strength to the fibrous matrix of the foam at this stage, the volume fraction is not sufficient to resist external damage. When the volume fraction of gelatin is 1.6%, there is no hollow or collapse phenomenon in the longitudinal direction, but the cellulose is messy. The results show that under these conditions, although the foam network structure can resist the damage during molding to a certain extent, the addition of gelatin is still very low. Therefore, the degree of physical crosslinking is insufficient, and the foam cannot retain not its complete network structure vertically. In contrast, when the volume fraction of gelatin is 2.4% or 3.2%, a significant change in the foam morphology can be seen in the surface SEM images. This is mainly due to the physical cross-linking between the gelatin and the cellulose. It can be seen from the cross-sectional SEM image that the same cell structure as that of the foam surface appears in the longitudinal direction, the cell wall is firm, and there is no bending phenomenon, indicating that a three-dimensional network structure is formed, which is beneficial to improve the mechanical strength of the foam. Among the samples, when the volume fraction of gelatin is as high as 3.2%, there is an excessive crosslinking in the foam. The cellulose connected by the second liquid is piled up to form lamellae; the interaction between the gelatin and the cellulose is hindered by the cross-linking effect between the gelatin and the gelatin molecules in lamellae [47]. This results in a stratification phenomenon, as shown in the figure in the longitudinal direction. Table 1 shows the density and porosity of the cellulose-based foam prepared under different gelatin concentrations, where the density is less than 0.01 g/cm^3^. At the same time, the porosity reaches more than 98%, meeting the requirements of ultra-lightweight and porous.

### 3.2. Pore Size Distribution of the Cellulose-Based Foam

An automatic mercury injection apparatus is used to characterize the pore size of cellulose-based foams with different gelatin concentrations. Figure 2a shows the mercury injection curve. In the low-pressure region, the mercury injection amount gradually increases, this stage mainly fills the large hole. With the increase in pressure, the amount of mercury injected is no longer increasing. At this time, the energy consumption is mainly used for the volumetric compression of the material [48]. It can be seen from the curve that the foam sample has a macroporous structure (having pores larger than 500 nm is called macroporous). When the volume fraction of gelatin is 0 or 1.2%, the constant value of the amount of mercury injected is the same. With the increase in the volume fraction of gelatin, the amount of mercury injected increases significantly. This indicates that when the volume fraction of gelatin is 0 or 1.2%, there are more large-aperture bubbles. The detailed pore size distribution is shown in Figure 2b. Without the addition of gelatin, the pore size distribution of the foam is uniform and large. After the addition of the gelatin, the pore size of the foam is significantly reduced. As the volume fraction of the gelatin increases, the pore size distribution curve of the foam exhibits a multimodal phenomenon. When the volume fraction of the gelatin is 2.4%, the one strong peak that appears in the large-size region, as in other concentrations, is accompanied by other small peaks that appear in the small-size region. This indicates the presence of a hierarchically interconnected porous structure, which is beneficial to improve the mechanical properties of the foam [49]. The detailed data are shown in Figure 2c. With the increase in the gelatin volume fraction, the pore size of the foam first decreases and then increases. When the volume fraction of the gelatin is 2.4%, the pore size of the foam is the smallest, and the average pore size is approximately 47 μm, which is 178 μm less than the pore size before adding gelatin, which was 225 μm. The results show that based on the temperature-controlling properties of gelatin, the addition of a second liquid can not only improve the strength of the cellulose skeleton but also refine the pore size of the foam.

In addition, it has been confirmed that maintaining the stability of the network through the “patch-bridge” mechanism only is not enough. By comparing the average pore diameter (225 μm) of the cellulose foam after curing without gelatin with the average bubble diameter (113 μm) of the wet foam under the same conditions, it is found that the bubbles not only become thicker but are also subjected to other strong destructive forces during the curing process. For example, in the process of freezing to generate the ice crystals, the force generated by the expansion of the growth volume of the ice crystals pushes the cellulose [50]. This results in the destruction of the cellulose network that is inherently unstable and has a larger pore size. In contrast, based on the temperature-controlled cross-linking properties of the gelatin, gelation occurs rapidly at low temperature after the addition of gelatin [51]. Moreover, the liquid bridges connected between the cellulose are converted into solid bridges, which allows the foam structure to be fixed before it can be coarsened. Not only is the original network structure of the wet foam maintained, but also the gelatin elastic aggregates tend to be connected through the forces of the polar group interactions such as hydrogen bonds and so on during the aging process. At the same time, the gelatin elastic aggregates are pulled into the overlapping distance that exists between the cellulose [52]. Therefore, it reduces the pore size of the foam. When the gelatin concentration is further increased to 3.2%, the liquid bridge endows too much force to the cellulose wet foam. The cross-linking between the gelatin and the gelatin molecules hinder the interaction between the gelatin and the cellulose. This results in the stratification of the foam, the destruction of network structure, and the increase in the foam pore size. The results show that the second liquid concentration has a significant effect on the pore size distribution of the cellulose-based foam. When the volume fraction of gelatin is 2.4%, the pore size of the cellulose-based foam reaches the minimum value.

### 3.3. Crystal Structure of the Cellulose-Based Foam

The crystal structure of the cellulose-based foam is studied and analyzed by an X-ray diffractometer. The addition of the amorphous gelatin can reduce the ordered structure of the foam to a certain extent. As shown in Figure 3a, the main diffraction peaks appear at 2θ = 16.5 and 22.5°, corresponding to the crystal planes (110) and (200), respectively. This figure conforms to the type I structure of typical cellulose [53]. For NPGB, the characteristic peaks are similar to NPB, indicating that the crystal structure of cellulose did not change after the addition of gelatin. When the volume fraction of gelatin is increased from 0% to 2.4%, the crystallinity of the foam decreases from 76.81% to 28.20%, and when the volume fraction of gelatin is further increased, the crystallinity (CrI) tends to increase too, as shown in Figure 3b. To analyze the reasons behind this pattern, the addition of amorphous gelatin may not only reduce the crystallinity of the foam but also contribute to the formation of the three-dimensional network structure of the cellulose-based foam [7]. With the addition of gelatin, the orientation of the cellulose that always tends to spread in a plane is partially broken, and the circular cell structure is gradually presented in the longitudinal direction. The foam converts from ordered to disordered, so the overall crystallinity decreases. When the volume fraction of the gelatin reaches 2.4%, the three-dimensional network structure of the cellulose-based foam is completely formed. At this stage, the disorder of the foam reaches its maximum and the crystallinity of the foam reaches its minimum. When the concentration of the gelatin further increases, the foam undergoes delamination in the longitudinal direction and the three-dimensional structure is damaged. Thus, the crystallinity of the foam increases and its disorder decreases.

### 3.4. Mechanical Properties of the Cellulose-Based Foam

The stress-strain curve can reflect the change in the internal structure of the foam under the action of a force and the foam’s response to it [54]. From the stress–strain curve shown in Figure 4, it can be seen that there are two distinct response regions. However, Chen and Kobayashi et al. [55] suggest the presence of three areas, in which the elastic deformation of the pore wall and the compression of the macropores mainly occur in the linear elastic area, the plastic yield of the pore wall occurs in the platform area, and the densification of the foam porous structure often occurs in the densification area. According to the literature [56], this is related to the processing route of the materials. For example, the lightweight foam which Cervin et al. [8] prepared through Pickering foam does not have a platform area, and when the strain reaches 60%, it directly switches from the linear elastic area to the dense area.

In this study, the same behavior was observed in the above-mentioned report. Moreover, before the deformation reaches 60%, the stress increases linearly with the strain in the linear elastic region. Then, the stress extends to the densified area, and no platform appears. From the curves in Figure 4, it can be seen that the stress of the solid foam increases at first, and then decreases under the same deformation conditions with the increase in gelatin volume fraction in the linear elastic region. The stress reaches the maximum value when the gelatin volume fraction is 2.4%. The compressive stress values at 60% strain for NPB, NPGB-1.2, NPGB-1.6, NPGB-2.4, and NPGB-3.2 are 1.35546 kPa, 24.13049 kPa, 34.23665 kPa, 55.74601 kPa, and 54.50891 kPa, respectively. Furthermore, at 80% strain, the respective stress values reach 14.03921 kPa, 116.98721 kPa, 148.10305 kPa, 289.73374 kPa, and 216.98205 kPa. These are higher than the previously reported values for cellulose foam/aerogel shown in Table 2. When the volume fraction of gelatin increases up to 3.2%, the stress decreases slightly. This may be caused by the cross-linking of the excessive gelatin, which results in a vertical stratification. This destroys the three-dimensional network of cellulose-based foam, leading to stress concentration and decreased mechanical strength.

### 3.5. Adsorption and Cycle Performance of the Cellulose-Based Foam

The driving force of the superhydrophobic/superoleophilic cellulose foam in absorbing oil is derived from the capillary force of the superhydrophobic hierarchical structures [64]. As shown in Figure 5a, the highly porous cellulose-based foams have an absorption capacity of 20 to 60 g/g for the various organic solvents or oils. They mainly absorb the liquid through the capillary action and physical sealing in the narrow spaces of the pores. Due to the loose structure and the difficulty in forming the gelatin cellulose-based foam (NPB), this chapter only investigates the adsorption capacity of gelatin cellulose-based foam (NPGB). By comparing the absorption capacities of NPGB-1.2, NPGB-1.6, NPGB-2.4, and NPGB-3.2 for different solvents, it is found that when the volume fraction of gelatin is 1.2%, 1.6%, or 2.4%, there is no significant difference in the absorption capacity of the foam, and the maximum absorption capacities are 52.040 g/g, 55.625 g/g, and 57.061 g/g, respectively. In contrast, when the volume fraction of gelatin reaches 3.2%, the absorption capacity of the foam reduced significantly, and the maximum absorption capacity is 31.606 g/g, which may be due to the damage of the three-dimensional structure of the foam.

The traditional extrusion–drying–absorption cycle method can be used to test the cycle performance of the cellulose-based foam. The specific process is shown in Figure 5b. A certain volume of the foam is placed in different organic solvents/oils. After adsorption, the adsorbed solvent is extruded by simple extrusion and the foam is soaked in ethanol overnight for desorption. The foam is then transferred to a vacuum drying oven and dried at 60 °C for 10 h. This process is called a cycle. The cellulose-based foam is a kind of porous adsorption material which plays an important role in the recycling of materials and the adsorption of substances.

Figure 5c shows the absorption efficiency of a cellulose-based foam containing different gelatin volume fractions at different cycles, using chloroform as an example. It can be seen that the foam undergoes one adsorption, and the absorption amount decreases by 20–65% during the next adsorption. After five adsorption cycles, the adsorption capacity decreased by 73%. This may be analyzed by two factors. On the first hand, in the first adsorption process of the foam, the organic solvents and the oil enter the foam holes and stay there causing a blockage, so they become difficult to completely desorb. Therefore, no additional liquid storage space remains for the next adsorption. On the other hand, when desorption occurs after the last adsorption is complete, although a large amount of the solvent adsorbed in the foam can be discharged by the extrusion method, the structure of the foam is severely damaged. Thus, it results in a decrease in the adsorption capacity.

By comparing the circulating capacities of the foams with different gelatin concentrations, we found that the adsorption capacity of the foam cycle when the volume fraction of the gelatin is 2.4% or 3.2% is worse than that when the volume fraction is 1.2% or 1.6%. The reason is analyzed according to the surface morphology and the size distribution of the foam. The addition of the gelatin can make the cell wall of the cellulose-based foam thicker and the cell smaller. Where the mechanical strength and the water retention capacity of the foam are improved. After one round of adsorption, it is more difficult to desorb the oil droplets from the foam, and the recovery capacity of the foam decreases. The 3.2% gelatin volume fraction is the worst among the tested volume fractions in terms of liquid absorption and circulation. The results show that the gelatin concentration has an important effect on the adsorption and the recyclability of the cellulose-based foam. As the gelatin concentration increases, the adsorption cycle capacity decreases.

## 4. Conclusions

In this study, BF, NFC, TAD, and GAC are used as the raw materials, and SDS is used as a green solvent to prepare the cellulose-based foams with different gelatin concentrations via capillary foam technology. The results show that the structure of cellulose-based foam after adding gelatin is obviously compact, the crystallinity is decreased, the disorder is increased, and the mechanical properties are improved, but the recycling ability is decreased. Furthermore, the results show that the mechanical properties of the cellulose-based foam are improved, but its circulation capacity is decreased. The mechanical properties of the cellulose-based foam improve with the increase in the gelatin volume fraction. However, excessive addition of gelatin causes excessive crosslinking of foam, which destroys the three-dimensional network of the cellulose-based foam, leading to stress concentration and decrease in the mechanical strength, and the experimental results show that when the volume fraction of gelatin is 2.4%, the stress is 55.746 kPa. At this time, the mechanical properties of the cellulose-based foam are the best, and the adsorption capacity of the foam also reaches the highest 57.061 g/g. In addition, with the increase in gelatin concentration, the circulation capacity of the cellulose-based foam decreases, and the number of adsorption cycles is limited, among which, the adsorption capacity decreased by 73% after five adsorption cycles. In addition, with the increase in gelatin concentration, the adsorption and circulation capacity of gelatin decreases. Therefore, the concentration of the secondary liquid (gelatin) has a great influence on the mechanical properties as well as the adsorption and circulation capacity of the cellulose-based foam.

## Figures and Tables

**Figure 1 polymers-15-01879-f001:**
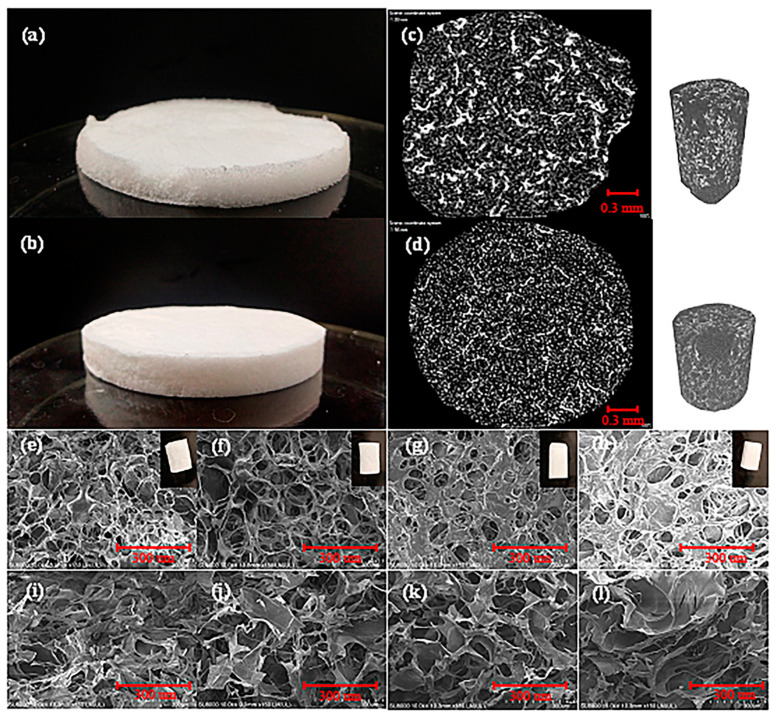
(**a**) Physical appearance of NPB and NPGB foams; (**b**) X-ray diffraction profile; and (**c**,**d**) surface (**e**–**h**) and section (**i**–**l**) SEM images of the cellulose-based foam at different gelatin volume fractions (from the left to the right, they are NPGB-1.2, NPGB-1.6, NPGB-2.4, and NPGB-3.2).

**Figure 2 polymers-15-01879-f002:**
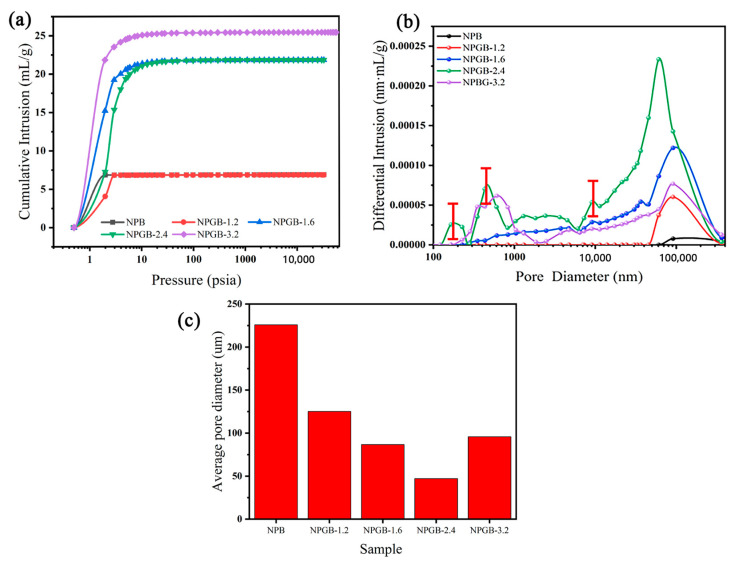
(**a**) Mercury injection curve, (**b**) pore size distribution, and (**c**) average pore size of cellulose-based foam with different gelatin volume fractions.

**Figure 3 polymers-15-01879-f003:**
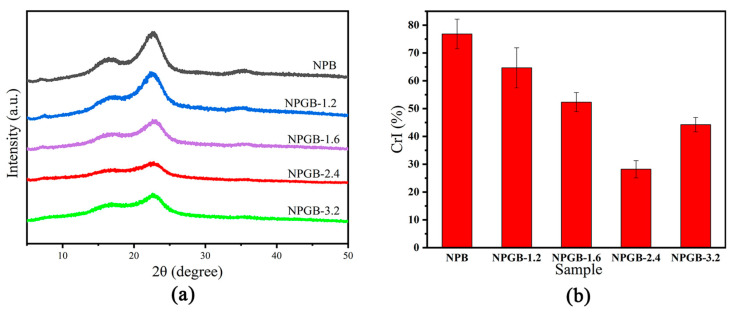
(**a**) XRD spectra and (**b**) crystallinity index (CrI) of cellulose-based foam with different gelatin volume fractions.

**Figure 4 polymers-15-01879-f004:**
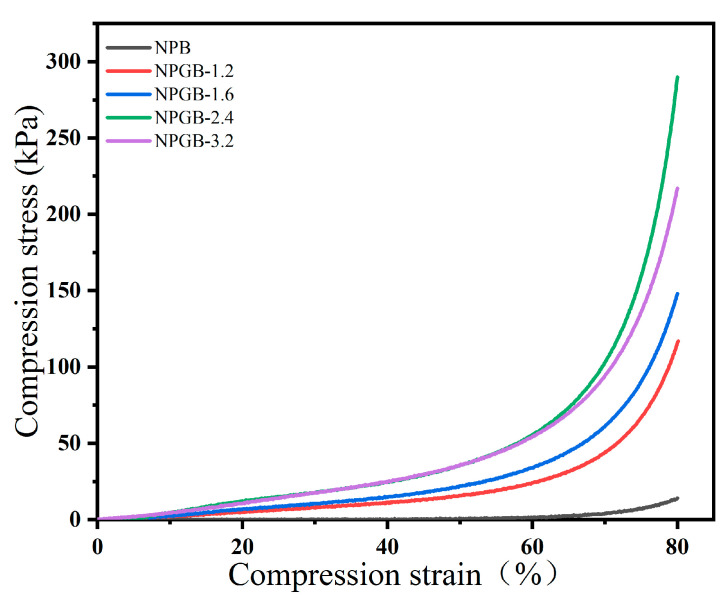
Stress–strain curve of cellulose-based foam with the different gelatin volume fractions.

**Figure 5 polymers-15-01879-f005:**
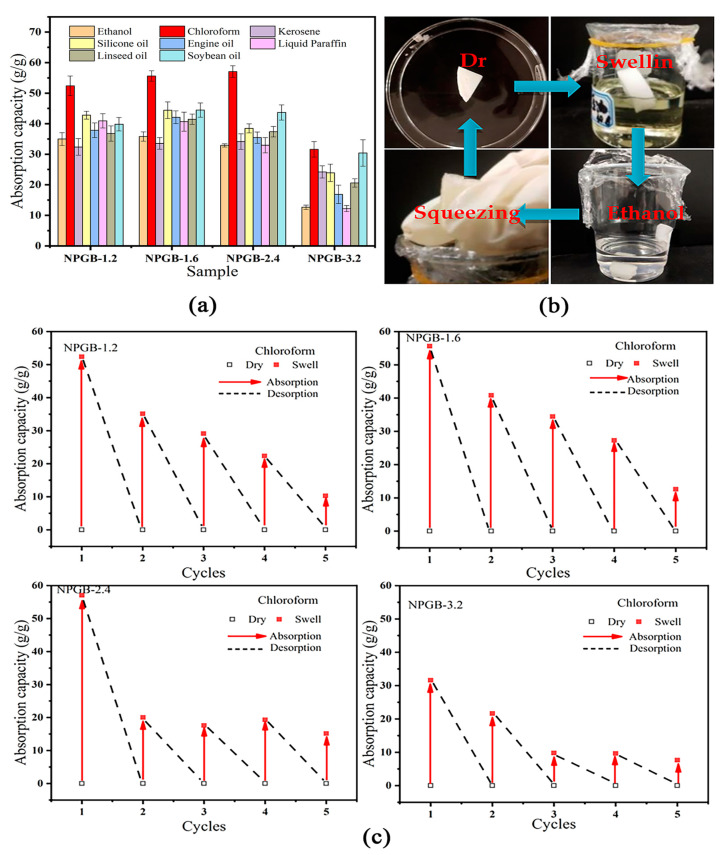
(**a**) Absorption capacity of foam for different organic solvents/oils, (**b**) the flowchart of the cycle for adsorption–desorption, and (**c**) the adsorption capacity of the foam for different organic solvents/oils at different gelatin volume fractions.

**Table 1 polymers-15-01879-t001:** Physical parameters of the cellulose-based foam at different gelatin volume fractions.

Sample	Density (g/cm^3^)	Porosity (%)
NPB	0.0108 ± 0.0012	99.30 ± 0.05
NPGB-1.2	0.0155 ± 0.0030	98.95 ± 0.15
NPGB-1.6	0.0145 ± 0.0050	99.03 ± 0.05
NPGB-2.4	0.0175 ± 0.0050	98.84 ± 0.03
NPGB-3.2	0.0173 ± 0.0034	98.86 ± 0.23

**Table 2 polymers-15-01879-t002:** Summary of the properties of the physical parameters of the cellulose-based foam materials.

Material	Density (g/cm^3^)	Porosity (%)	Mechanical Strength (kPa)	References
Novel cellulose foam	0.096–0.0175	≥98%	55.746 (60%)	
NFC/MFC foam	0.010–0.060	90.0	13.78	[29]
NFC aerogels	<0.030	99.7	13.78	[57]
Silanized NFC sponge	0.017	99.0	27.70 (50%)	[58]
NFC foam	0.010	99.4	12.00 (50%)	[59]
Lignin/cellulose	0.010	80.0–90.0	200.00	[60]
Cellulose foam	0.020–0.065	80.0–90.0	10.00–90.00	[61]
Nano cellulose foam	0.011	97.1–99.4	60.00 (80%)	[62]
Cellulose scaffold	0.006–0.176	99.7	271.00 (70–80%)	[63]

## Data Availability

No new data were created or analyzed in this study. Data sharing is not applicable to this article.

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
