# Peer review of "Green Preparation of Lightweight, High-Strength Cellulose-Based Foam and Evaluation of Its Adsorption Properties"

_polymers, 2023, doi:10.3390/polym15081879_

Round 1

Reviewer 1 Report

This article is devoted to the production of cellulose-based foam, as well as the evaluation of its adsorption properties. The article in terms of volume and subject matter meets the requirements of the Journal. The topic presented by the authors seems to be relevant, since the direction of obtaining functional materials based on natural substances has been actively developing recently. In addition, adsorbents obtained on the basis of these materials are also in demand in various fields. Despite the many advantages of this article, there are some points that you need to pay attention to:

1. ABSTRACT needs to be expanded.

2. In the introduction it is necessary to add a paragraph about cellulose, what it is, where it is used (hydrolysis, hydrogenation, sulfation, etc.). It is desirable to quote at this point: 10.1007/s00226-022-01363-4.

3. It is advisable for the authors to double-check the quality of the English language and the terminology used.

4. The term "crystallinity index" is known from the literature, but I first encountered the term "crystallization index" in this article. Since the authors do not study the process of crystallization, it is desirable to double-check the appropriateness of the use of this term.

5. Each experimental data must be compared with literature sources. Thus, the dynamics of the studied processes becomes clear. I recommend that the authors carefully compare their findings with the literature.

6. XRD data should be compared with other pulp modification processes. For example, with sulfation. Recently, Polymers published an article to which you can refer.

7. Adsorption performance. Why do the authors use this model for research? In addition, it is interesting to consider the kinetics and thermodynamics of adsorption of this material. However, this may be the subject of further research.

8. Conclusions can also be expanded.

Author Response

We feel very sorry for the careless uploading error of the last Word file. And we have uploaded the correct file again. Please see the attachment. 

Reviewer 2 Report

Current manuscript entitled “Green preparation of lightweight, high-strength cellulose-1 based foam and evaluation of its adsorption properties” by “Zhou et al” deliberated on the structural stability of the solid foam by adding a secondary liquid via the capillary foam technology. The effects of the addition of different gelatin concentrations on the micro-morphology, crystal structure, mechanical properties, adsorption, and recyclability of the cellulose-based foam are investigated. Manuscript seems good and can be accepted after addressing the following comments.

1.      Remove the highlights.

2.      Enhance the quality of Graphical Abstract.

3.      Abstract is not constructive enough. Please revise.

4.      Clear statements of the novelty of the work should also appear briefly in the Abstract and Conclusions sections.

5.      The paper should be carefully revised for punctuation, grammar, spelling mistakes and sentences structuring.

6.      Improve the image quality of the figures.

7.      Revise the figure 1.

8.      Scale bar is not clear for the SEM images.

9.      Authors should mention about the following articles related to green synthesis. Agro-waste to sustainable energy: A green strategy of converting agricultural waste to nano-enabled energy applications. Waste-to-energy: Utilization of recycled waste materials to fabricate triboelectric nanogenerator for mechanical energy harvesting. Flexible PVDF based piezoelectric nanogenerators. Cellulose an ageless renewable green nanomaterial for medical applications: An overview of ionic liquids in extraction, separation and dissolution of cellulose. Acoustic-electric conversion and triboelectric properties of nature-driven CF-CNT based triboelectric nanogenerator for mechanical and sound energy harvesting.

10.  In the introduction last paragraph mention what is lacking in the literature and what has been done in the current work.

Round 2

Reviewer 1 Report

Accepted

Reviewer 2 Report

Aceept